# Identification of Aquatic Organisms Using a Magneto-Optical Element

**DOI:** 10.3390/s19153254

**Published:** 2019-07-24

**Authors:** Kohei Oguma, Tasuku Sato, Tomohiro Kawahara, Yoshikazu Haramoto, Yoko Yamanishi

**Affiliations:** 1Department of Mechanical Engineering, Kyushu University, 744 Motooka, Nishi-ku, Fukuoka-shi, Fukuoka 819-0395, Japan; 2Department of Biological Functions Engineering, Kyushu Institute of Technology, 2-4 Hibikino, Wakamatsu-ku, Kitakyushu 808-0196, Japan; 3Department of Life Science and Biotechnology, National Institute of Advanced Industrial Science and Technology (AIST), 1-1-1 Umezono, Tsukuba 305-8555, Japan

**Keywords:** magneto-optics, MEMS, magnetism

## Abstract

In recent advanced information society, it is important to individually identify products or living organisms automatically and quickly. However, with the current identifying technology such as RFID tag or biometrics, it is difficult to apply to amphibians such as frogs or newts because of its size, stability, weakness under a wet environment and so on. Thus, this research aims to establish a system that can trace small amphibians easily even in a wet environment and keep stable sensing for a long time. The magnetism was employed for identification because it was less influenced by water for a long time. Here, a novel magnetization-free micro-magnetic tag is proposed and fabricated with low cost for installation to a living target sensed by Magneto-Optical sensor for high throughput sensing. The sensing ability of the proposed method, which was evaluated by image analysis, indicated that it was less than half of the target value (1 mm) both in the water and air. The FEM analysis showed that it is approximately twice the actual identification ability under ideal conditions, which suggests that the actual sensing ability can be extended by further improvement of the sensing system. The developed magnetization-free micro-magnetic tag can contribute to keep up the increasing demand to identify a number of samples under a wet environment especially with the development of gene technology.

## 1. Introduction

As sensing tracing technology advances, the demand to automatically trace large numbers of objects moving at higher speeds is increasing. In the field of biology, examples include cell tracking [1,2,3] to track cell dynamics; engraftment and differentiation to evaluate the behaviors and functions of cell groups; and tagging [4] to label, release, and then re-capture fish and shellfish. Tagging has the potential to estimate the amount of fish and shellfish resources as well as reveal their migration ranges. 

Typical examples of automatic recognition technology currently in widespread use include two-dimensional symbols such as QR (quick response) codes [5], RFID (radio frequency identification) tags [6,7], and biometrics technology. It is said that the smaller the size, the higher the cost and the more difficult it is to introduce to small living organisms. An RFID tag, which is a card or a rod-like tag including a circuit, is used in a wide variety of transmission/reception rewritable tags. In recent years, a one tag per one product system has been introduced in convenience stores for product identification, and unmanned operation has been achieved by reading the product information and processing the purchase at the cash register [8]. The RFID communicates between the transmitter and the tag via electromagnetic waves. The distance that electromagnetic waves can travel is limited by the path loss. In aqueous environments, electromagnetic waves are easily absorbed by water molecules, resulting in a short communicable distance [7,9]. 

Biometrics technology uses biological information unique to each individual, such as retinal blood vessels, fingerprints, and irises possessed by living organisms for identification [10,11,12], and reads such information with a special device such as a camera or infrared light. To read biological information, a high-accuracy sensor is often required. These devices tend to be large and expensive, but are highly safe because the information cannot be copied to other individuals. Since this technology needs to identify the target site for recognition, the target must be authenticated voluntarily. Although it can be used for human identification, it is difficult to use for animals that have difficulty communicating with people. Especially, identification of aquatic organisms is difficult with currently available technology.

In this paper, we aim to establish a system that can trace small amphibians easily in the wet environment stably for a long time. We strive to develop magnetic tags and sensing systems because a magnetic field is less affected by an aqueous environment than electromagnetic waves. A magnetic field can be easily measured in one-dimension by a magnetoresistance effect element that utilizes the change in electric resistance of the magneto-impedance element/solid state using the fact that the impedance is sensitively changed by an external magnetic field due to the skin effect of the Hall element/high-magnetic permeability alloy of a magnetic body using an electromotive force in a direction orthogonal to both the current and magnetic fields. However, in order to obtain magnetic information in many places or planar information, an array of numerous elements or scanning by moving the elements is required. Hence, the system is expensive and identification is complicated [13]. 

A two-dimensional field sensing method using the relation between magnetism and optics has been proposed. Herein, a magneto-optical (MO) sensor is adopted as it can provide a planar image, which represents the strength of the magnetic field with a resolution of about 10 μm in a single element via optical rotation of polarized light in a magnetic field and with a light cutoff. Moreover, because two-dimensional magnetic field sensing eliminates the need for measurements at multiple locations, in short, rapid automatic identification can be achieved. There are other advantages, including favorable sensitivity with different material of the element, low cost, stability at room temperature, polarity discrimination, and good responsiveness for dynamic observation. Studying this rotation of polarized light provides insights into the magnetic behavior of materials that have a potential to minimize the magnetic recording and the improvement of processing speeds in the future [14]. The further improvement of magnetic devices for higher density and speed in data storage demands an examination of their magnetic domains [15]. Generally, MO sensors have been employed to inspect surface cracks on aircraft fuselages. Specifically, they have been used to detect leaked magnetic fluxes generated near cracks, which is useful to rapidly inspect for corrosion and subsurface cracks [16]. However, they have also disadvantages of magnetic hysteresis, because they use magnetic materials in the element, difficulty to quantify the magnetic force, resolution limit of its magnetic domain size. 

The present paper is configured as follows. In Section 2, we describe the method of individual identification using an MO sensor and magneto-optics as well as the relation between a magnetic field and polarization. We compare sensing distance measurements using an MO sensor and FEM analysis in Section 3. In Section 4, we discuss the results, while we provide conclusions in Section 5.

## 2. Materials and Methods

### 2.1. Magneto-Optical Sensor

Light in which the vibration direction is aligned in one direction is called polarization, and a surface that vibrates is called a polarization surface. When light passes through a substance to which a magnetic field is applied, a phenomenon occurs in which the polarization plane rotates according to the amount of magnetization of the substance. This phenomenon is called the Faraday effect. 

The Faraday rotation angle is expressed as [17,18]
*θ_F_* = V H L(1)

Here, *θ_F_* is the Faraday rotation angle, L is the sample thickness, H is the applied magnetic field intensity, and V is the Verdet constant, which determines the material-dependent ease of rotation. The MO sensor uses the Faraday effect. It is composed of a light source, a CCD camera, a magneto-optical element, and a polarizer/analyzer. When light from the light source passes through the polarizer, it is polarized and directed to the element. The polarization plane rotates in accordance with the magnetic field intensity. Transmission/quenching in accordance with the angle is performed by the analyzer, and the magnetic field intensity can be converted to a light-and-dark image for presentation. The brightness *I_D_* after cutoff is expressed as
*I_D_* = *I*_0_ cos^2^(*θ_p_* + *θ_F_* − *θ_A_*)(2)

Here, *I*_0_ is the light intensity of the incident light, *θ_p_* is the angle of the polarizer, *θ_F_* is the Faraday rotation angle, and *θ_A_* is the analyzer angle. The magneto-optical element is magnetized by the magnetic field to be sensed, and the rotation angle is determined based on the direction of the spin of the magnetic domain, which is the minimum unit of magnetization. Since the effect of the magnetic field becomes smaller the further it is from the observation target, it is necessary to make the magneto-optical element as thin as possible and to observe the effect close to the target (Figure 1). However, as the thickness of the magneto-optical element decreases, the rotation angle decreases. Consequently, the obtained contrast decreases. Iron yttrium garnet is used as the main magneto-optical element material since a large rotation angle can be obtained with a small thickness by using a material with a large Verdet constant such as ferrimagnetic metal like iron. In addition, the Magneto-Optical element has anisotropy, that is to say, there is specificity in the rotation angle, and whether the detection film has an in-plane, inclined, or vertical orientation depends on the application.

In this experiment, we used a mag-eye powered by a Dino-Lite digital microscope premier (A-type, AnMo Electronics Corp., New Taipei, Taiwan) as a magneto-optical sensor, which responds to only the magnetic field incident in the direction perpendicular to the element surface (Table 1 and Table 2). The north pole was brighter and the south pole was darker. We captured the image by image analysis software (Dino Capture 2.0, AnMo Electronics Corp., New Taipei, Taiwan). Detailed specification of the sensor is shown in Table 1, and the set-up is summarized in Table 2. 

### 2.2. Identification Method

In this research, we developed a micromagnetic tag that can identify biological information in aqueous environments over a long time. Figure 2 shows the sensing concept. Since the distance between the sensor and the tag affects the change in the magnetic field around the magnetic substance, we set the parameter as the sensing distance D. Biometric information, which was expressed by patterning a magnetic material, was finely fabricated by SU-8 photolithography. The tags were divided into identification information sections represented by squares and position identification marker sections. After creating a tag, it was introduced subcutaneously in a living organism. The tags were read for the presence of a magnetic material above the skin by means of an MO sensor that visualized the vertical magnetic flux. In this study, a coil or a magnet was placed outside and a static magnetic field was applied to strengthen the magnetic field of the tag. It is not necessary to magnetize the magnetic substance in advance using this method. However, the advantage is that the risks of demagnetization, which may occur over time, and a thermal disturbance of the performance of the magnet, can be eliminated by the fact that the tags are magnetized each time they are sensed.

## 3. Results

### 3.1. Creating Micromagnetic Tags

The response to a magnetic field or the signals may depend on the type of magnetic substance used for the tag material. To select the appropriate magnetic substance to seal the magnetic tag as magnetic information, we tested three types of materials: (**a**) nickel as a soft magnetic material, (**b**) iron powder as a soft magnetic material (50 μm or less in diameter), and (**c**) gold as a diamagnetic material. In the tests, all materials responded to the magnetic field. In the manufacturing process of each tag, a concave surface was formed by photolithography, and patterning was performed by plating or etching (Figure 3). The thin film of PDMS (polydimethylsiloxane) was cured with the temperature of 90 °C. Au sputtering was performed at about 300 mA (**a**) for 100 s (**c**) for 5 min using a Magnetron sputtering device (MSP-30T, vacuum device co., Inc., Ibaraki, Japan). The photosensitive resin SU-8 3050 (micro chem Corp., Westborough, MA, USA) used in (**a**) and (**b**) was applied to the substrate so that the thickness can be about 200 μm, and wet etching was carried out by immersing in Propylene Glycol Methyl Ether Acetate (kanto chemical co., Inc., Tokyo, Japan) for 15 min at room temperature. We baked SU-8 layer for 90 min as pre bake and for 15 min as post exposure bake at 90 °C. In the process of (**b**), iron powder was mixed with PDMS for molding and placed in a depression to replace the magnetic material to create a tag (Figure 4). Since iron powder sinks in uncured PDMS, we increased the density as much as possible in this experiment, and by mixing well at the time of preparation and quick curing, substantially uniform magnetic properties were obtained as an aggregate. In the process of (**c**), in order to help OFPR-800LB (tokyo ohka kogyo co., Ltd., Kanagawa, Japan) sticking to the Au layer, we applied OAP (Oligomeric Adhesion Promoter, tokyo ohka kogyo co., Ltd., Kanagawa, Japan) on the substrate and baked it for 2 min at 90 °C. We baked the OFPR layer for 30 min as pre bake and for 10 min as post exposure baked at 90 °C. Also for the development, we soaked the substrate in NMD-3 (Tetramethylammonium hydroxide 2.38%, tokyo ohka kogyo co., Ltd., Kanagawa, Japan) for 10 min at room temperature, and soaked it in a gold etchant (AURUM-302, kanto chemical co., Inc., Tokyo, Japan) for 1 min. The rest of OFPR was removed by acetone (99.5%, kanto chemical co., inc., Tokyo, Japan). Then, the MO-sensing suitability of each tag was evaluated.

### 3.2. Magnetic Field Conditions

Figure 5 shows the experimental setup. Varying the direction of the external magnetic field changes the magnetization direction of the tag. The direction and number of magnetic fluxes incident onto the element are changed, affecting the recognizable sensing distance. We performed experiments for an in-plane magnetic field and an off-plane magnetic field, which are a magnetic field parallel and perpendicular to the magneto-optical element, respectively (Figure 6). The sensing distance D, which is the distance between the tag and the sensor, was observed under the most easily observable condition: D = 0 μm. 

Ferrite magnets were exposed on both sides of the tag to apply a magnetic field, and tag imaging was performed (Figure 7). For the off-plane magnetic field, the external magnetic field itself is sensed. Hence, the magnetic substance cannot be strongly magnetized. The number of magnetic fluxes decreases and the image obtained changes as the magnetization direction changes. The in-plane magnetic field is better suited for tag recognition since the contrast of the tag information section is higher and more clearly recognizable for the in-plane magnetic field than for the off-plane one. Moreover, although the tag using gold is hardly recognizable due to the smaller magnetization amount, the tag using iron powder or nickel is recognizable due to the sufficient magnetization of the magnetic substance. From the viewpoint of biocompatibility, we decided to use a tag containing iron powder since a living organism may react negatively to nickel. Also, this tag can be stably identified half a year after being produced, so they are stable for a long time.

### 3.3. Sensing Distance Measurements

When the tag is embedded under the skin of a living organism, the sensing distance D corresponds to the thickness of the skin. If the living organism has thick skin, the magnetic field information of the tag becomes difficult to read due to the diffusion of the magnetic flux. We set the target value of sensing distance as 1 mm considering 0.6 mm is the maximum thickness of Xenopus, a kind of frog [19]. Hence, we evaluated the effect of skin thickness when a tag is actually implanted under the skin and an in-plane magnetic field was applied to the iron powder tag. We quantified the brightness of the S-pole portion of the magnetic substance of the photographed image by image analysis software and illustrated the relation between the distance and the brightness. We measured the luminance six times for each point and showed the reference luminance, which is the lowest luminance of the image without a tag by a dotted line for comparison of luminance. 

Figure 8 and Figure 9 show the results of evaluating the limit values of sensing distance D in air and water. As the sensing distance increases, the effect of the tag magnetic field is reduced due to spreading of the magnetic flux, and the contrast of the section becomes low. When the sensing distance is D = 70 μm or less, the change in the sensor’s luminance is negligible showing around the lower limit of 60 μm. As the sensing distance D increases, the luminance increases, making identification difficult. The sensing distance is limited to about D = 230 μm, provided that sensing is possible below the reference luminance.

### 3.4. FEM Analysis

In order to improve the sensing distance, it is conceivable to expand and improve the detection ability of the sensor, to strengthen the magnetic flux and the magnetic field, and to reduce spreading of the magnetic flux. FEM analysis can estimate the sensing distance in the case where the precision of the magneto-optical sensor increases because the increase in contrast and the reduction of the mechanical and optical loss are achieved. FEM not only allows a magnetic substance other than iron to be evaluated, but also allows the tag shape to be changed in a simple manner. A tag image similar to that of the MO sensor can be obtained by converting only the magnetic flux incident in the vertical direction of the measurement surface into a monochrome and displaying it using FEM analysis software. The vertical magnetic field sensing range was set to ± 2 mT like the MO sensor. For comparison to the results in Section 3.3, we determined the sensing limit of the iron powder tag when the in-plane magnetic field was applied from the luminance of the photographed image. 

The tag visibility is improved by expanding the range of luminance, which affects the contrast of the tag (Figure 10). As the sensing range increases, the luminance approaches 135, which is the intermediate value, due to diffusion of the magnetic flux (Figure 11). Since FEM analysis does not reveal a variation in luminance, the sensing distance limit is 460 μm, assuming that the distinguishable luminance lower limit value is 127 from the lower limit value of the luminance range of an actual MO sensor. This is a distance twice as large as the sensing limit obtained by the actual MO sensor, indicating that the sensing distance can be extended by the development of the MO sensor and changing the magnetic substance.

## 4. Discussion

Among the tags using nickel, iron, and gold, only the gold one is difficult to sense. The magnetic susceptibility of each material, which represents the ratio of magnetization to the applied magnetic field, seems to determine the sensing ability. Gold is a diamagnetic material, and its magnetic susceptibility has a negative value. We considered that sufficient magnetization is not obtained for sensing because its absolute value is about 10^8^ smaller than that of soft magnetic materials such as nickel and iron [20]. If a diamagnetic material is magnetized to the same extent as nickel, sensing could be possible. However, a diamagnetic substance that generates a demagnetizing field in itself that can cancel or overwhelm the external magnetic field does not exist. Consequently, it is optimal to use a soft magnetic substance.

The end of the section appears blurry in the iron tag compared to the nickel tag. This is attributed to the distance between the tag and the sensor as an offset due to the shrinkage of PDMS during the creation and the lack of iron powder filling completely to the upper end. Although PDMS are employed so as to connect iron powder, it is possible that the presence of PDMS among iron powder particles may deteriorate the magnetization performance. However, it was found that increasing the density of iron particles by adjusting the amount of PDMS does not degrade the sensing ability because the sensing ability is equivalent to that obtained when the sensing distance of a nickel tag was measured.

Two kinds of magnetic fields (in-plane magnetic field and off-plane magnetic field) were applied in a static situation as a magnetic field application direction. Regardless of the kind of field, the magnetic substance is magnetized to generate both magnetic poles in the direction of the magnetic field. The emitted magnetic flux is directed to the element enough for sensing. Since the off-plane magnetic field is perpendicular to the magneto-optical element, the element also senses an external magnetic field. Intensifying the external magnetic field for strong magnetization causes it to reach the upper sensing limit. Because the tag information is buried with the saturation by an external magnetic field, a strong magnetic field could not be applied. On the other hand, since the in-plane magnetic field is in a direction not sensed by the sensor, parallel direction, a strong magnetization could be achieved by applying larger magnetic field. The intensified magnetization affects the amount of the emitting magnetic flux, thus the elements become able to sense the information even in the distance, which consequently extends the sensing limit. In addition, for the in-plane magnetic field, both the N-pole and the S-pole of the magnetic substance appear in the observation plane. With the white and black part appearing on one screen, the entire sensing range is utilized enough to recognize the brightness difference of each section. Hence, even with the same amount of magnetization, the patterning of the magnetic substance can be relatively easily determined. 

In this article, we considered only the parallel arrangement of the MO element and the sample. If the MO element and the sample are placed at a large inclination to the tag, the distance between the end portions becomes large and they cannot be brought close to each other, so the sensitivity becomes worse. In the case of a slight inclination, the magnetic flux to the element is almost unchanged, so the sensitivity does not change.

In Section 3.3, we investigated whether the sensing ability exceeds the target distance which corresponds to the thickness of the skin when the tag is embedded in a living organism. The target value is defined as 1 mm taking a sufficient margin. The sensing distance limit by the MO sensor is 200 to 250 μm in both air and in water, indicating that the sensing system is not susceptible to aqueous environments. However, increasing the sensing distance causes the luminance to gradually rise. In both air and water, the value changes by 10 because the luminance is changed due to the daily conditions of the element and the camera in the sensor and cooling by the presence of liquid around it. This does not have a large influence on the sensing ability. In addition, the results of FEM analysis indicate the sensing ability is nearly 460 μm, which is twice the MO measurement. This difference is mainly due to the difference between the tag material and the sensing performance. For the tag mixed with PDMS and iron powder embedded in the section, in FEM analysis, one section represents one iron plate, and the fineness of the magnetization impacts the direction of the magnetic flux and the formation of the magnetic field such as whether each iron powder is magnetized or one iron plate is magnetized. Since the iron powder concentration is as high as possible, it is thought that the iron powders are almost in contact with each other, but do not reach the magnetic properties of the iron plate. The sensing distance corresponds to the skin thickness between the tag and the sensor when implanted in a living organism. A larger sensing distance is required for use in various living organisms, however, the sensing ability did not reach the in-vitro target value of 1 mm, and its value remained at 230 μm, which is half of FEM results. These results suggest that the sensing ability can be improved and the sensing distance can be extended by improving the sensor, tag material or fabrication process, and imaging process. The materials such as thin magneto-optical elements with a large turning angle [21] and high resolution may improve the sensor’s ability and imaging process by using a background subtraction process or a noise reduction process during MO sensing by convoluted integration through time-lapse image superposition, which may achieve a clear tag image in the distance. This sensing system can be used widely for insects and animals unless they have hair or thick skin because they make larger sensing distance. When used mainly for amphibians, this system exerts an advantage of being resistant to water.

## 5. Conclusions

In this research, we developed a magnetic tag that is less susceptible to an aqueous environment using a magnetic substance inside living organisms and established a system to quickly read the information transmitted by the tag. After installing the magnetic tag fabricated by photolithography to the targeted amphibians, we read the identifying information through an MO sensor assisted with the external magnetic field. Although the tag magnetic substance and external magnetic field configurations were optimized, we could not reach the target value when the sensing ability was evaluated from both the MO sensor and FEM analysis. Compared with these two experiments, FEM analysis showed that the sensing ability is twice the actual measurement mainly due to the difference between the tag material and the sensor’s performance, which should be improved in the future. This sensing system can be used widely for insects and animals, especially for amphibians. This system exerts an advantage of being resistant to water. We believe that this research will contribute to tracing technologies by loading reagents and genetic information introduced when embedded in a living organism.

## Figures and Tables

**Figure 1 sensors-19-03254-f001:**
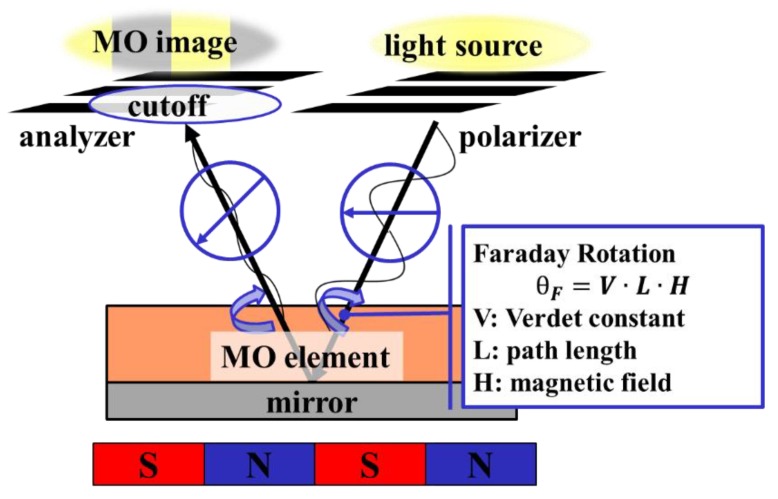
Principle of the magneto-optical sensor, which visualizes the magnetic flux density. Rotation angle of linearly polarized light is proportional to the strength of the magnetic field. Rotated polarized light is cutoff by an analyzer according to its angle.

**Figure 2 sensors-19-03254-f002:**
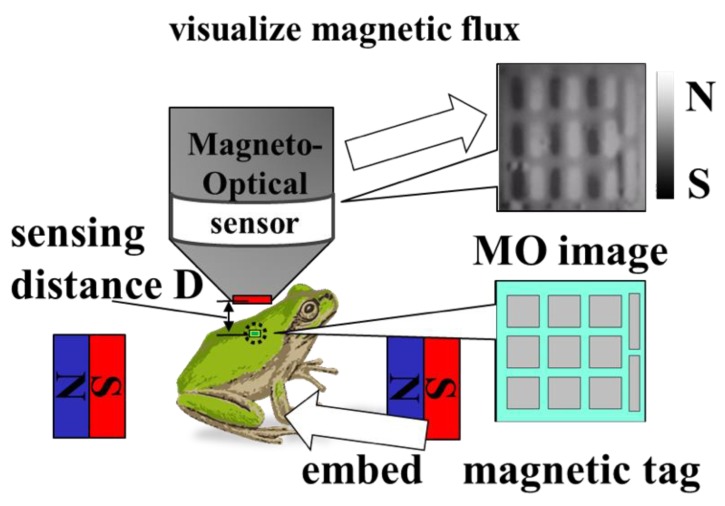
Concept of a magnetization-free magnetic tag by applying an external magnetic field to a magnetic tag and sensing by the MO sensor.

**Figure 3 sensors-19-03254-f003:**
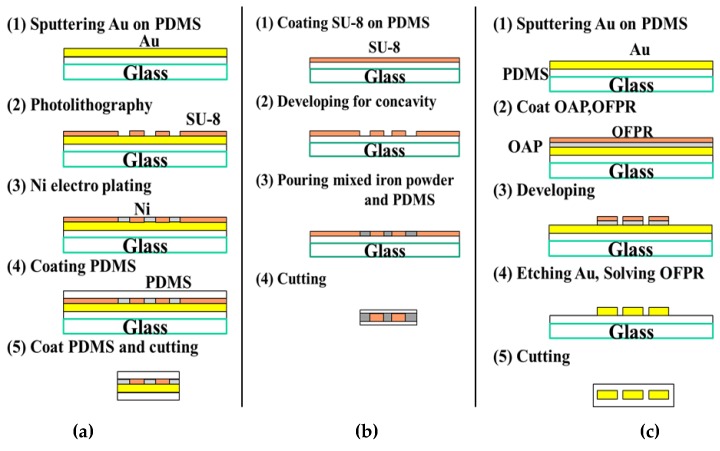
Detailed magnetic tag creation process. (**a**) Nickel plating is selectively performed by forming SU-8 insulation in the conducting part made by gold sputtering, and a pattern carrying information is produced. (**b**) Mixture of PDMS and iron powder is poured into the depression of SU-8 made by photolithography. (**c**) Tag-type photolithography is applied to gold sputtered on a substrate. Exposed gold is dissolved by immersion in the gold etchant, leaving a tag-shaped gold thin film.

**Figure 4 sensors-19-03254-f004:**
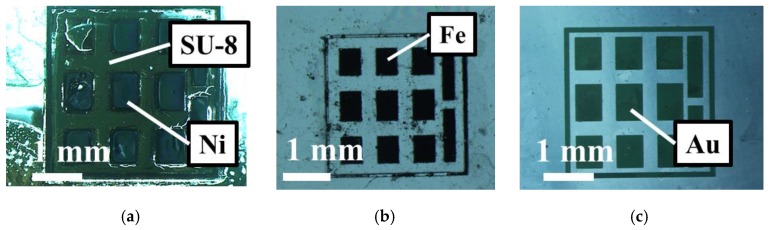
Magnetic tag using different materials. (**a**) Tag created by nickel electrolytic plating. Upper left section is a section without nickel because it is not plated well. (**b**) Tag created using PDMS and iron powder. Iron powder is sufficiently contained in each section. (**c**) Tag created by gold etching.

**Figure 5 sensors-19-03254-f005:**
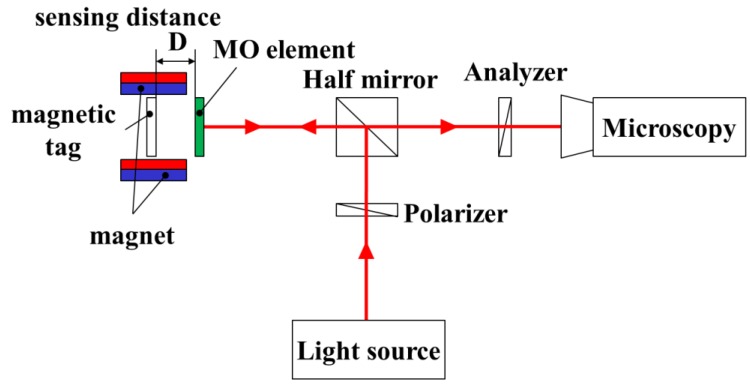
Experimental setup. After polarized light from the light source is rotated by the magneto-optical element, the analyzer cuts off the light according to its angle. Half mirror has a role of passing through half of the irradiated light and reflecting the rest. External magnetic field is controlled by changing the magnet arrangement around the tag.

**Figure 6 sensors-19-03254-f006:**
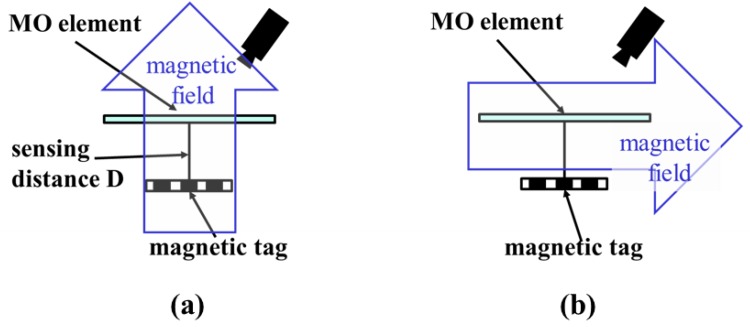
Two conditions of the magnetic field direction. (**a**) Magnets are placed opposite to the top and bottom of the off-plane magnetic field tag, and a magnetic field is applied perpendicular to the tag and the magneto-optical element. (**b**) Magnets are exposed opposite on the left and right of the in-plane magnetic field tag, and a magnetic field is applied in parallel with the tag and magneto-optical element.

**Figure 7 sensors-19-03254-f007:**
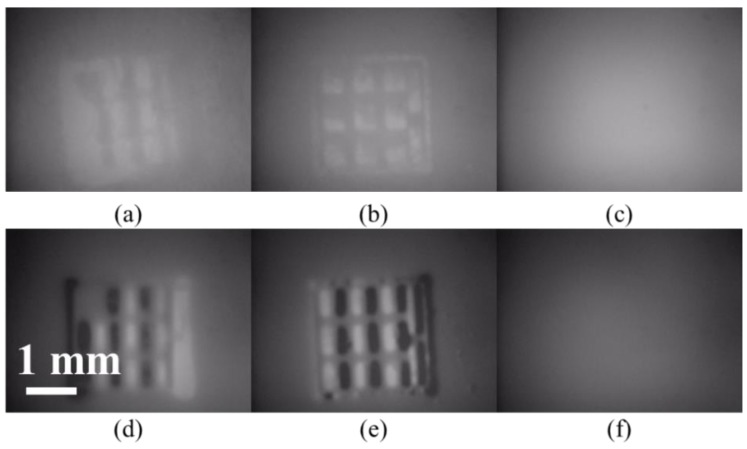
(**a**–**c**) Off-plane magnetic field to nickel, iron and gold tags, respectively. (**d**–**f**) In-plane magnetic field to nickel, iron and gold tags, respectively. Gold tag has a small magnetization and tag identification is difficult, but the conditions of (**d**,**e**) are the most suitable for tag identification.

**Figure 8 sensors-19-03254-f008:**
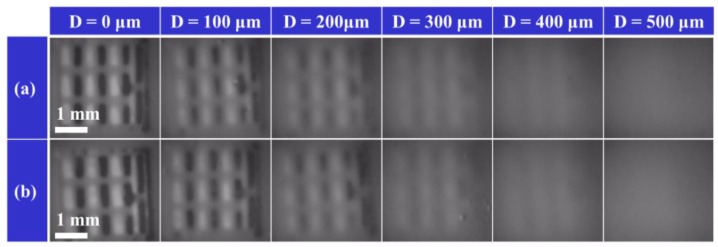
Change in the tag photographed image due to a change in sensing distance. (**a**) Sensing image in air. (**b**) Sensing image in water. Both are less than 300 μm, making it possible to identify tag sections with the naked eye.

**Figure 9 sensors-19-03254-f009:**
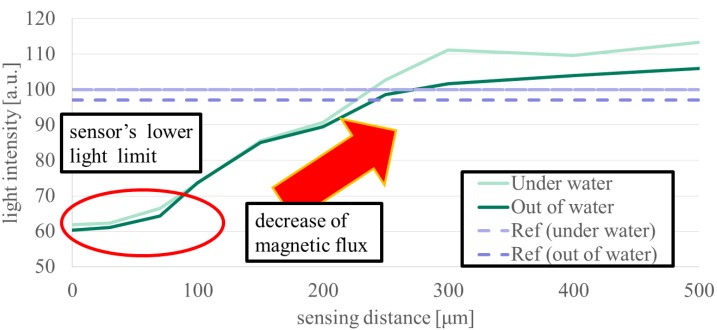
Relation between MO sensing distance and brightness by image analysis software. As the sensing distance D increases, the brightness approaches a constant value, and eventually tag identification becomes difficult. Sensing ability does not significantly differ in air or water. Tag identification is possible in the range of less than 250 μm.

**Figure 10 sensors-19-03254-f010:**
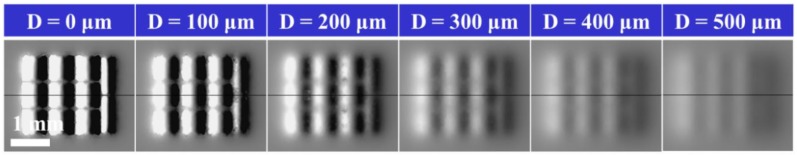
Sensing distance by FEM analysis software. Tag identification is possible even at 400 μm, which is difficult to identify with an actual MO sensor under ideal conditions.

**Figure 11 sensors-19-03254-f011:**
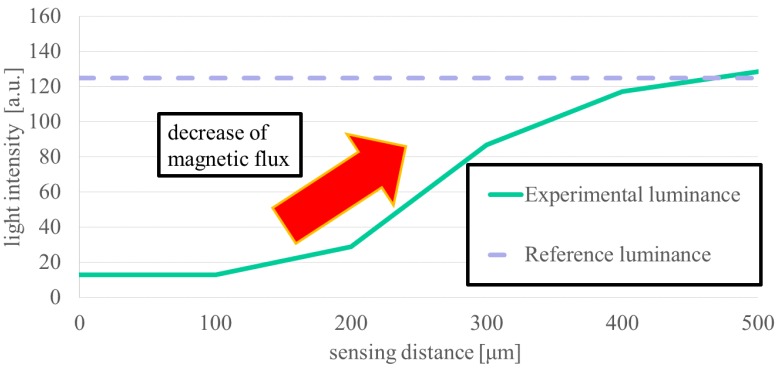
Relation between sensing distance and luminance by FEM analysis. The sensing ability has been reinforced and the limit is extended to around 500 μm.

**Table 1 sensors-19-03254-t001:** Specifications of Magneto-Optical sensor used in this experiment. It has enough resolution to read the information of the tag.

Term	Specification
Sensor specification	Mag-view, type-A
Element resolution	25 [µm]
Camera resolution	5M [px]
Measuring range	±2 [mT]
Field of view	0.8 x 0.8 [mm]
Frame rate	Up to 30 [fps]
Size	dia.3 x 11 [cm]

**Table 2 sensors-19-03254-t002:** Detailed camera settings of Dino Capture 2.0. In order to stabilize the image, we utilized the monochrome photographs.

Term	Specification
Brightness	128
Contrast	16
Hue	0
White balance	0
Saturation	16
Sharpness	1
Gamma	64
Others	monochrome

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
