# Peer review of "Identification of Aquatic Organisms Using a Magneto-Optical Element"

_sensors, 2019, doi:10.3390/s19153254_

Reviewer 1 Report

This manuscript presents experimental study the aim of which is to identity living organisms in water with tools of magneto-optics. It sounds very interesting, important, and useful. However, I cannot recommend this manuscript for publication due to following reason:

1)   The manuscript is written absolutely unclear. It is not clear what the authors supposed to measure: Concentration of micro-flora in water solution or the chemical composition of the fluid inside living macro-organisms (like a frog shown in Fig. 2).

2)  Have the authors already performed their measurements or are they just going to complete it?

3)  There are several physical mistakes in the manuscript.

If and when the authors will revise their manuscript, I will agree to review it again.

Additional comments:

1)  Faraday rotation [as well as Eq. (1)] is true only if the magnetic field is perpendicular to the surface. If the magnetic field is in-plane of surface this situation is called Voigt configuration and is described by other equations [see for example Phys. Rev. B 89,125312 (2014) ]

2)  Light incidence should be near-perpendicular to the surface.

3)  If light is reflected from mirror (as shown in Fig. 1), then its phase can loss (in some situation)  pi/2 (please check whether this is correct in the considered situation). Light reflects also from the surface of MO element (see Fig. 1), not only from the mirror. The phenomenon of rotation of light polarization which reflects from the surface is called Kerr magneto-optical rotation.

4) The first two sentences in Sec. II are written as for schoolchildren but not in scientific style.

 Author Response

Thank you very much for reviewing our paper. Please see the answers corresponding to your comments.  

1)   The manuscript is written absolutely unclear. It is not clear what the authors supposed to measure: Concentration of micro-flora in water solution or the chemical composition of the fluid inside living macro-organisms (like a frog shown in Fig. 2).

- Thank you very much for your valuable comment. We have revised the manuscript to clarify the target of this research.

 2)  Have the authors already performed their measurements or are they just going to complete it?

-Thank you for your comment. The in-vitro measurement in the experiment is now over, however the installing experiment has not been done yet. Perform in-vivo experiments after achieving in-vitro stage targets.

 3)  There are several physical mistakes in the manuscript.

-Thanks a lot. We have revised the manuscript.

 If and when the authors will revise their manuscript, I will agree to review it again.

 Additional comments:

 1)  Faraday rotation [as well as Eq. (1)] is true only if the magnetic field is perpendicular to the surface. If the magnetic field is in-plane of surface this situation is called Voigt configuration and is described by other equations [see for example Phys. Rev. B 89,125312 (2014) ]

-Thank you for your valuable comment. Certainly, in the case of an in-plane magnetic field, it becomes a Voigt arrangement, and Cotton-Mouton effect occurs depending on the magnetic field strength, and it causes birefringence. However, birefringence is unnecessary to explain this sensor principle. Although it is possible that the image appears double due to the Voigt effect, this element is more responsive to the vertical magnetic field, and the thickness is sufficiently thin so that the effect of birefringence is almost zero as we confirmed. In other words, I think it is not necessarily described because there is a risk of confusing the reader by describing the fork placement and principle.

 2)  Light incidence should be near-perpendicular to the surface.

-Thank you for your comment. The light may be actually tilted a little inside the sensor. As for the sensor principle, this display format is used to describe it in an easy-to-understand manner without losing its physical meaning. The sensor's internal structure, Figure 5, has been modified to be vertically incident.

 3)  If light is reflected from mirror (as shown in Fig. 1), then its phase can loss (in some situation)  pi/2 (please check whether this is correct in the considered situation). Light reflects also from the surface of MO element (see Fig. 1), not only from the mirror. The phenomenon of rotation of light polarization which reflects from the surface is called Kerr magneto-optical rotation.

- Thank you for your comment. Certainly, the light reflected from the garnet surface, which is the material of the element, rotates under the Kerr effect. However, most of the light is transmitted without being reflected on the surface of the element, so the effect can be said to be sufficiently smaller than the Faraday effect. Although it is useful because magnetic domain observation using the Kerr effect, etc., it is not described in this paper because it is not the intended principle and the effect is very small.

 4) The first two sentences in Sec. II are written as for schoolchildren but not in scientific style.

- Thank you. We have revised that sentence.

Reviewer 2 Report

This manuscript presents a micromagnetic tag using a magnetic substance inside living organisms, which allows a system to read the information transmitted by the tag. Authors optimize the external magnetic field and detection ability through FEM analysis. However, this manuscript must be improved considering the following comments:
1. -The redaction of the abstract section is confusing. This section must consider the innovation or scientific contribution and the main advantages and results.

2.-Introduction section needs more information about the main scientific contribution and advantages of the proposed sensors with respect to others reported in the literature. In addition, authors must add the main challenges of the proposed sensors.

3.-Section of materials and methods. This section requieres more detail technical information about the operation principle and proposed methodology. For instance, data of the technical parameters of the sensors, dimensions, and experimental setup.

4.-Results. This section must contain more information about the used parameters in the fabrication process (e.g., temperature, chemical substances, etching techniques). In aditions, authors must indicate more discussions about the results reported in Figures 7-11. Authors must mention the main challenges of the proposed methodology.

5.-Conclusions section must contain more information of the proposed method, the main results and advantages.Author Response

 Thank you very much for your reviewing comment. Please find out attached our answers corresponding to your comments. 

1. -The redaction of the abstract section is confusing. This section must consider the innovation or scientific contribution and the main advantages and results.

-Thank you for your comment. Abstract has been modified to clarify the benefits and results of this study.

 2. -Introduction section needs more information about the main scientific contribution and advantages of the proposed sensors with respect to others reported in the literature. In addition, authors must add the main challenges of the proposed sensors.

-Thank you for your comment. The advantages are it is difficult to add detailed information because there are several types of MO sensors. We added defects and scientific contributions of the sensor.

 3.-Section of materials and methods. This section requieres more detail technical information about the operation principle and proposed methodology. For instance, data of the technical parameters of the sensors, dimensions, and experimental setup.

-Thank you for your comment. We added tables on sensor sensing range and resolution, fps, and image acquisition settings.

 4.-Results. This section must contain more information about the used parameters in the fabrication process (e.g., temperature, chemical substances, etching techniques). In aditions, authors must indicate more discussions about the results reported in Figures 7-11. Authors must mention the main challenges of the proposed methodology.

-Thank you for your comment. Detailed information on photolithography (types of SU-8 and NMD-3, pre-bake time temperature, etc.) was added. We added discussions about the results The issues of the proposed method are described in the first two sentences of 3.3, but we added the comparison of the results with the issues (target values) in the discussion.

 5.-Conclusions section must contain more information of the proposed method, the main results and advantages.

-tThank you for your comment. We added descriptions of methods and results.

Reviewer 3 Report

p.p1 {margin: 0.0px 0.0px 0.0px 0.0px; font: 12.0px Helvetica} p.p2 {margin: 0.0px 0.0px 0.0px 0.0px; font: 12.0px Helvetica; min-height: 14.0px}

Comments and suggestions for authors:

The authors present an analysis of a micro-magnetic tag using s magneto-optical element as a sensor for small living organisms. It is an interesting idea, and they show the potential capabilities of this approach under different experimental conditions. The main result in the current manuscript is the proof of concept for future application. However, they conclude that additional optimization is required to achieve the application stage. 

The manuscript is well written, and it has a sequential order to present the main ideas. I recommend to publish the paper after minor revisions, listed below:

1. The experiments configuration is limited to a parallel arrangement of the MO-element and the sample. How does the sensitivity change with a tilted configuration?

2. Based on the experimental results, the range with good sensitivity is in the order of 100 microns. In a potential application, how are you going to ensure that the magnetic tag and the magnetic field are collinear? 

3. How are the magnetic properties of the skin compared with the current experimental conditions used in the present manuscript (air and water)?  

4. What kind of animals are you targeting with this study? It could be interesting to discuss the skin thickness of different animals, and the surface patterns, roughness, or spikes over it. Those features can limit the application of the sensor. 

5. Is the iron powder stable on the PDMS? Is there any evidence of aggregates? If so, how do they affect the sensitivity of the magnetic tag?

6. What is the target value of the sensor? You mention that on the conclusions. But, it is not evident anywhere in the manuscript.

7. Line 163. I believe there is a typo: “off”-plane magnetic field instead of “in”-plane magnetic field. 

Author Response

Thank you very much for your reviewing comment. Please find out our answers corresponding to your comment shown below.

1. The experiments configuration is limited to a parallel arrangement of the MO-element and the sample. How does the sensitivity change with a tilted configuration?

-Thank you for your comment. If the MO element and the sample are placed at a large inclination to the tag, the distance between the end portions becomes large and cannot be brought close to each other, so the sensitivity becomes worse. In the case of a slight inclination, the magnetic flux to the element is almost unchanged, so the sensitivity does not change.

 2. Based on the experimental results, the range with good sensitivity is in the order of 100 microns. In a potential application, how are you going to ensure that the magnetic tag and the magnetic field are collinear?

-Thank you for your comment. A slight tilting of the external magnetic field has little effect on discrimination and does not have to be perfectly parallel. I think that it is enough to be aware that two magnets are placed on the side of the object with the N pole and the S pole facing each other, and that a parallel magnetic field is applied to some extent.

 3. How are the magnetic properties of the skin compared with the current experimental conditions used in the present manuscript (air and water)? 

-Thank you for your comment. Unfortunately, the magnetic properties of the skin are unknown. The static magnetic field is basically not affected by anything but magnetic materials, so it is probably not inhibited by the skin. In addition, most of the frog's skin is composed of protein, so if you compare air and water protein should be close to water. We want to investigate it in the future.

 4. What kind of animals are you targeting with this study? It could be interesting to discuss the skin thickness of different animals, and the surface patterns, roughness, or spikes over it. Those features can limit the application of the sensor.

- Thank you for your valuable comment. The subject is limited to the subjects with thin skin and no hair, but this research can be used widely for insects and animals. And when used mainly for amphibians, it exerts an advantage of being resistant to water.

→ Added to confusion and discussion.

 5. Is the iron powder stable on the PDMS? Is there any evidence of aggregates? If so, how do they affect the sensitivity of the magnetic tag?

- Thank you for your valuable comment. Iron powder is stable even half a year after it is made. The settling speed of iron powder is sufficiently slow compared to the curing time of PDMS, and the magnetic properties of the mixture as a dispersion of iron powder are the almost uniform by mixing well at the time of preparation. When iron particles are in contact with each other, they can be regarded as larger magnetic bodies by forming NS poles between the particles in contact with each other, which may improve the magnetic properties. We want to investigate it in the future.

We added that the tag is stable even after half a year.

 6. What is the target value of the sensor? You mention that on the conclusions. But, it is not evident anywhere in the manuscript.

- Thank you for your valuable comment. I’m afraid although not written in the manuscript, the target value of sensing distance D is 1 mm.

Thus, it can be said that sufficient performance can be achieved if there is a sensing distance of about 0.6 mm. We aim to put it to practical use as a 1 mm target for larger sizes.

[In reference 18] Xenopus, a kind of frog we're targeting, the total thickness of the skin ranges from 0.15 to 0.20 mm in males and from 0.38 to 0.54 mm in females.

→ Appended with reference.

 7. Line 163. I believe there is a typo: “off”-plane magnetic field instead of “in”-plane magnetic field.

-Thank you, you're right. We corrected a typo of that sentence.

Round  2

Reviewer 1 Report

I have already reviewed this manuscript. The authors  took into account part of my comments and I can recommend now the manuscript for publication.

Author Response

Thank you very much for your comment 「I have already reviewed this manuscript. The authors  took into account part of my comments and I can recommend now the manuscript for publication.

Thank you again for reviewing our manuscript.  

Reviewer 2 Report

Authors have improved their manuscript considering the reviewer's comments. However, authors must include more recent references (e.g., references between 2015-2019) related with micro-magnetic tags and magneto-optical sensors.

Author Response

Thank you very much for your comments. We have added recent references [14][21] to the manuscript (green marking part).
